# Modulating Nucleus Oxygen Concentration by Altering Intramembrane Cholesterol Levels: Creating Hypoxic Nucleus in Oxic Conditions

**DOI:** 10.3390/ijms23095077

**Published:** 2022-05-03

**Authors:** Joao Seco, Clarence C. King, Gianmarco Camazzola, Jeannette Jansen, Luca Tirinato, Maria G. Marafioti, Rachel Hanley, Francesca Pagliari, Scott P. Beckman

**Affiliations:** 1Division of Biomedical Physics in Radiation Oncology, DKFZ German Cancer Research Center, 69120 Heidelberg, Germany; gianmarco.camazzola@gmail.com (G.C.); j.jansen@dkfz-heidelberg.de (J.J.); tirinato@unicz.it (L.T.); mariagrazia.marafioti@gmail.com (M.G.M.); rachel.hanley3@mail.dcu.ie (R.H.); f.pagliari@dkfz-heidelberg.de (F.P.); 2Department of Physics and Astronomy, Heidelberg University, 69120 Heidelberg, Germany; 3School of Mechanical and Materials Engineering, Washington State University, Pullman, WA 99164, USA; clarence.king@wsu.edu (C.C.K.); scott.beckman@wsu.edu (S.P.B.)

**Keywords:** nucleus oxygen concentration, cholesterol, OER, hypoxia, HIF, Warburg Effect

## Abstract

We propose a novel mechanism by which cancer cells can modulate the oxygen concentration within the nucleus, potentially creating low nuclear oxygen conditions without the need of an hypoxic micro-environment and suited for allowing cancer cells to resist chemo- and radio-therapy. The cells ability to alter intra-cellular oxygen conditions depends on the amount of cholesterol present within the cellular membranes, where high levels of cholesterol can yield rigid membranes that slow oxygen diffusion. The proposed mechanism centers on the competition between (1) the diffusion of oxygen within the cell and across cellular membranes that replenishes any consumed oxygen and (2) the consumption of oxygen in the mitochondria, peroxisomes, endoplasmic reticulum (ER), etc. The novelty of our work centers around the assumption that the cholesterol content of a membrane can affect the oxygen diffusion across the membrane, reducing the cell ability to replenish the oxygen consumed within the cell. For these conditions, the effective diffusion rate of oxygen becomes of the same order as the oxygen consumption rate, allowing the cell to reduce the oxygen concentration of the nucleus, with implications to the Warburg Effect. The cellular and nucleus oxygen content is indirectly evaluated experimentally for bladder (T24) cancer cells and during the cell cycle, where the cells are initially synchronized using hydroxeaurea (HU) at the late G1-phase/early S-phase. The analysis of cellular and nucleus oxygen concentration during cell cycle is performed via (i) RT-qPCR gene analysis of hypoxia inducible transcription factors (HIF) and prolyl hydroxylases (PHD) and (ii) radiation clonogenic assay every 2 h, after release from synchronization. The HIF/PHD genes allowed us to correlate cellular oxygen with oxygen concentration in the nucleus that is obtained from the cells radiation response, where the amount DNA damage due to radiation is directly related to the amount of oxygen present in the nucleus. We demonstrate that during the S-phase cells can become hypoxic in the late S-phase/early G2-phase and therefore the radiation resistance increases 2- to 3-fold.

## 1. Introduction

Cellular membranes main biological function is the creation of a barrier between intracellular and extracellular environments as well as between different cellular compartments. This is largely due to the hydrophobicity of the membrane interior. Water soluble components such as ions and sugars can cross biological membranes through channels or transporters. Nonelectrolytes, like molecular oxygen, can cross the lipid bilayer membrane component by passive diffusion, permeating directly through the bilayer. Oxygen diffusion in cells and across membranes is of vital importance, as it allows the organelles (mitochondria, nucleus, etc.) to receive the oxygen that they need to survive and function. As a very small, nonpolar molecule, oxygen is generally thought to diffuse readily across cellular membranes. However, biological, biophysical and computational work strongly suggest that cholesterol can reduce membrane permeability to oxygen [1,2,3,4,5]. As a consequence, the cholestrerol presence within the membrane can alter there permeability to oxygen.

Extra-cellular plasma membranes or nuclear membranes are generally more highly enriched in cholesterol than other bilayers such as from organelles [6,7]. For cholesterol content between 0 and 15%, the membrane phospholipids behave like a liquid-disordered phase, which is typified by a more loose packing and less rigid nature of the phospholipids giving the membrane high permeability to oxygen diffusion. However, an enrichment in cholesterol (to 20–60%) causes an increase in membrane packing and the phospholipids enter a liquid-ordered phase, where they harbor a significant amount of cholesterol and have an increasing rigid nature depending on the packing density. The observed higher amount of membrane cholesterol yields a less permeable membrane and as a consequence there is a slower diffusion of oxygen across the membrane [1,2,4,5,8,9,10,11,12]. Novel cancer therapy approaches are being developed that target the lipid membrane of cancer cells, to allow improvement in chemotherapy, radiotherapy or immunotherapy effectiveness [12].

In the present work we propose a novel molecular mechanism by which cells can modulate the oxygen concentration within the nucleus and potentially create an hypoxic nucleus in the presence of external oxic conditions and without the need of a surrounding hypoxic microenvironment. The cellular ability to alter intra-cellular oxygen conditions depends on the amount of cholesterol present within the cellular membranes such as ER membranes or the plasma membrane. The mechanism centers on the competition between

**oxygen consumption** by the mitochondria, ER, golgi apparatus, etc, which removes oxygen. The oxygen consumption within the cytosol is primarily due to mitochondria. While within the ER and Golgi apparatus the oxygen removal is due to the formation of disulphide bonds needed for protein folding [13,14,15,16].**oxygen diffusion** within the cell, which replenishes consumed oxygen but is significantly affected by the cholesterol content of the various cellular membranes.

We use a systems theory approach to evaluate the inter-dependency of the extra-cellular oxygen and the intra-cellular oxygen diffusion and consumption, and how this may allow the cell to generate low oxygen conditions within the nucleus. Mathematically, these reaction-diffusion problems for oxygen have been studied in the literature [17,18,19,20,21], for oxygen consumption governed by nonlinear Michaelis-Menten reaction rates.

For the majority of the cellular conditions the oxygen diffusion rate is significantly greater than the oxygen consumption rate, yielding a surrounding medium that functions as a stable reservoir of oxygen because of the faster diffusion process. However, in the case of the nucleus, it is surrounded by the nuclear membrane, the cytosol (with mitochondria), the endoplasmic reticulum (ER) and Golgi apparatus. All these organelles contribute to the significant removal of oxygen and are composed of a significant number of membranes, that are semi-permeable to oxygen depending on the cholesterol content [22,23,24,25,26]. While the cholesterol levels in the ER/Golgi apparatus are lower than in the plasma membrane and nuclear membrane, the cumulative effect of multiple layers can still have a significant impact on oxygen concentration within the nucleus. In addition, the plasma membrane and nuclear membranes have usually high levels of cholesterol in cancer cells, thus reducing even more the amount of oxygen that reaches the nucleus. For a cell, the direction of oxygen diffusion proceeds from outside the cell, extracellular matrix (ECM), to cytosol, and from cytosol to the nucleus by crossing first the plasma membrane, then multiple membranes of the ER and finally the nuclear membrane. All membranes can function as semipermeable barriers to the diffusion of oxygen, effectively slowing the diffusion process.

The novelty of our work centers on the assumptions (i) that the many ER membranes, the plasma membrane and nuclear membrane can significantly lower the effective diffusion rate of oxygen within the cell and (ii) that oxygen removal becomes more effective for low values of the Michaelis constant, $K_{M}$, that characterizes the rate of oxygen removal via a Michaelis-Menten process. In the case of cancer cells or fast growing cells such as the Chinese hamster ovary (CHO) the $K_{M}$ is below 1% atm, while for the majority of the human healthy cells $K_{M}$ it is in the range 3–15% [27]. CHO is a healthy cell line that is the preferred cell line for radiation biology studies as they are fast growing cells and can be readily transfected for radiation biology studies, which were established in 1957 by Puck and colleagues from ovarian biopsy of an adult Chinese hamster [28]. For $K_{M}$ values below 1% atm, the consumption rate of the oxygen becomes of the same order as the slower diffusion rate of oxygen (due to cholesterol), allowing the cell to modulate the concentration of oxygen within the nucleus. In addition, cholesterol metabolism is dysregulated in many malignancies, including myeloid leukemia, lung, and breast cancers [29,30,31,32]. The impact of cholesterol on cellular function and cancer has been discussed in recent paper by Casares et al. [33], where membrane cholesterol content can have a major impact on the success of chemotherapy drugs such as cisplatin [34]. In Angles et al. [35] they demonstrated that membrane cholesterol affects significantly the oxygen transport and its availability on the cellular level for biological processes. The complexity of biological membranes is such that cholesterol is inhomogeneously distributed within the membrane and plays a substantial role in raft formation and stabilization. Lateral inhomogeneity of cholesterol content likely impacts oxygen transport, giving rise to regions of greater and lesser permeability.

## 2. Materials and Methods

### 2.1. Relative Oxygen Permeability of Membranes with Varying Cholesterol Content

Cholesterol increases the rigidity of the lipid bilayer and therefore decreases its permeability to oxygen, diminishing the volume of oxygen that can diffuse through the membrane per unit time (Table 1). In Table 1, the membrane permeability (in units of cm/s) is presented for different levels of cholesterol content within the membrane (data taken from [5]). The results indicate that the highest permeability is achieved for 0% cholesterol content, while a 20% reduction is observed when cholesterol content is 50%. In the case of ER membrane, cholesterol content within the membrane has been shown to be in the order of 5–10%, we thus estimated the permeability based on an average cholesterol content of 7.5% [24,25]. In addition, we present the relative permeability of each membrane relative to the case of 0% cholesterol.

### 2.2. Reaction-Diffusion Model of Oxygen Consumption within the Cell

In what follows a model is presented, that features the oxygen reaction-diffusion equations for the three main cellular regions: (i) the cytosol (which includes mitochondria), (ii) the endoplasmic reticulum (which includes smooth and rough ER and Golgi apparatus) and (iii) the nucleus, all represented in Figure 1.

The oxygen concentration in each region is assumed to reach steady state very quickly, allowing us to make the approximation $\frac{\partial O_{2}}{\partial t}=0$. We assume in this model that all oxygen consumption follows a Michaelis-Menten behavior, where for the case of the mitochondria the oxygen consumption was considered part of the cytosol. Mitochondria oxygen consumption follows the Michaelis-Menten behaviour as observed by studies of Chance and co-workers [36,37,38]. We also model the diffusion through a membrane that separates any two of the previous regions. The membrane is considered to be semi-permeable to oxygen, governed by a relative permeability parameter, **ϵ** that is the ratio between the permeability values of a membrane containing cholesterol and one without cholesterol. In Table 1 we provide the relative permeability ϵ value for the different regions within the cell. For the sake of simplicity, our model assumes that the cell is spherical, which allows the radial coordinate, r, to be the sole geometric descriptor. In the model, we assume that oxygen is diffusing from the extracellular matrix (ECM), through the cytosol, crossing the ER before reaching the nucleus, as represented in Figure 2.

The spherical geometry assumption for the cell allows us to solve numerically the nonlinear 3-dimensional problem of oxygen diffusion and consumption using the method developed by King (2019) [39]. The steady state reaction-diffusion equations are the following (in spherical coordinates):**Cytosol Diffusion and Consumption**
(1)$$D\left(\frac{2}{r}\frac{dO_{2}\left(r\right)}{dr}+\frac{d^{2}O_{2}\left(r\right)}{dr^{2}}\right)-\frac{V_{max,cyto}O_{2}\left(r\right)}{K_{M,cyto}+O_{2}\left(r\right)}=0,\;$$
**Endoplasmic Reticulum (ER) Diffusion and Consumption**
(2)$$D\left(\frac{2}{r}\frac{dO_{2}\left(r\right)}{dr}+\frac{d^{2}O_{2}\left(r\right)}{dr^{2}}\right)-\frac{V_{max,ER}O_{2}\left(r\right)}{K_{M,ER}+O_{2}\left(r\right)}=0,\;$$
**Nucleus Diffusion**
(3)$$D\left(\frac{2}{r}\frac{dO_{2}\left(r\right)}{dr}+\frac{d^{2}O_{2}\left(r\right)}{dr^{2}}\right)=0,\;\;$$
**Relative membrane Permeability (**ϵ**) and Diffusion**
(4)$$\epsilon D\left(\frac{2}{r}\frac{dO_{2}\left(r\right)}{dr}+\frac{d^{2}O_{2}\left(r\right)}{dr^{2}}\right)=0.\;\;\;$$

The extracellular matrix (ECM) is characterized by a constant concentration of oxygen, $C_{0}$. The oxygen will enter the cell by crossing the outer cell membrane. In the cell the oxygen will be characterized by diffusion (*D*, Fick’s first law) and consumption ($V_{max}$ and $K_{M}$, due to Michaelis-Menten kinetics) within the cytosol and ER regions, whereas in the nucleus we only assumed oxygen diffusion. The diffusion parameter **D** is considered to be the same in all regions, except within the membrane, where it is reduced due to the relative permeability **ϵ** of the membrane yielding a new diffusion: ϵD. The value of ϵ depends on which type of membrane and level cholesterol present in the membrane. The different types of membranes included are the Nuclear membrane ($\epsilon_{Nuc}$), the ER membrane ($\epsilon_{ER}$), and the outer Plasma Membrane ($\epsilon_{PM}$), as seen in Figure 2 and Table 1.

The Michaelis-Menten parameters **$V_{max}$** and **$K_{M}$** are considered to depend on the properties of oxygen sensing enzyme present within that region. Therefore we define different Michaelis-Menten parameters for the cytosol and ER, respectively (**$V_{max,cyto},K_{M,cyto}$**) and (**$V_{max,ER},K_{M,ER}$**).

Boundary conditions imposed are such that at each interface the oxygen concentrations and gradients are continuous and equal for each region either side of the interface.
(5)$$\left\{\begin{array}{c} \mathrm{Concentrations}:\;O_{2,j}\left(R_{j}\right)=O_{2,j-1}\left(R_{j}\right)\\ \mathrm{Gradients}\;(j\geq 1):\;{D_{j}\frac{dO_{2,j}\left(r\right)}{dr}|}_{r=R_{j}}={D_{j-1}\frac{dO_{2,j-1}\left(r\right)}{dr}|}_{r=R_{j}}\\ \mathrm{Gradient}\;(\mathrm{center}):\;{D_{center}\frac{dO_{2,center}\left(r\right)}{dr}|}_{r=0}=0 \end{array}\right.$$

Region **j = 0** corresponds to the extracellular matrix (ECM) surrounding the cell where $O_{2,0}\left(R_{1}\right)=C_{0}$ with $C_{0}$ constant oxygen concentration, $O_{2,j}$ is the oxygen concentration within region **j**, $R_{j}$ is the radial distance to interface between **j − 1** and **j**, $D_{j}$ is the effective diffusion for region **j**, which can be either D or ϵD. By definition of being a centrosymmetric system the derivative evaluated at r=0 must be 0.

### 2.3. Dimensionless Form of Reaction-Diffusion Model

Assuming Fickian diffusion and Michaelis–Menten consumption, the steady-state concentration in the cell, $O_{2}\left(r\right)$, is determined from the reaction-diffusion Equations (1)–(4), with boundary conditions given in Equation (Equation 5). This system of Equations (1)–(5) are transformed into the dimensionless form, using the following variable change [39].

Dimensionless radial position: $\rho =\frac{r}{R_{1}}$;Dimensionless oxygen Concentrations: $O_{2,j}^{D}\left(\rho \right)=\frac{O_{2,j}\left(r\right)}{C_{0}}$;Dimensionless Michaelis-Menten parameters:$\nu_{cyto}=\frac{V_{max,cyto}}{\left(D/R_{1}^{2}\right)*C_{0}}$, $\kappa_{cyto}=\frac{K_{M,cyto}}{C_{0}}$**and**$\nu_{ER}=\frac{V_{max,ER}}{\left(D/R_{1}^{2}\right)*C_{0}}$, $\kappa_{ER}=\frac{K_{M,ER}}{C_{0}}$.

Where $R_{1}$ is radial position of outer membrane of the cell at ECM interface, $C_{0}$ is constant external $O_{2}$ concentration for ECM and D is the global Fick’s diffusion constant. It is possible to further simplify the dimensionless form of Equations (1)–(5) by removing the linear term, $\frac{dO_{2}\left(r\right)}{dr}$, by performing the following substitutions: $u_{j}\left(\rho \right)=\rho O_{2,j}^{D}\left(\rho \right)$. Therefore the final expression for the reaction-diffusion equations that are implemented numerically are the following [39]:**Cytosol Diffusion and Consumption**
(6)$$\frac{d^{2}u_{j}\left(\rho \right)}{d\rho^{2}}-\frac{\nu_{cyto}u_{j}\left(\rho \right)\rho }{\kappa_{cyto}\rho +u_{j}\left(\rho \right)}=0,$$
**Endoplasmic Reticulum (ER) Diffusion and Consumption**
(7)$$\frac{d^{2}u_{j}\left(\rho \right)}{d\rho^{2}}-\frac{\nu_{ER}u_{j}\left(\rho \right)\rho }{\kappa_{ER}\rho +u_{j}\left(\rho \right)}=0,$$
**Nucleus Diffusion**
(8)$$\frac{d^{2}u_{j}\left(\rho \right)}{d\rho^{2}}=0,$$
**Relative membrane Permeability (**ϵ**) and Diffusion**
(9)$$\epsilon \;\frac{d^{2}u_{j}\left(\rho \right)}{d\rho^{2}}=0,$$ with boundary conditions given by:(10)$$u_{1}\left(1\right)=1\mathrm{and}u_{j+1}\left(\frac{R_{j+1}}{R_{1}}\right)=u_{j}\left(\frac{R_{j+1}}{R_{1}}\right),$$
(11)$$\begin{array}{cc} {\frac{du_{j}\left(\rho \right)}{d\rho }|}_{\rho =\frac{R_{j+1}}{R_{1}}}= & {\frac{1}{\epsilon }\frac{du_{j+1}\left(\rho \right)}{d\rho }|}_{\rho =\frac{R_{j+1}}{R_{1}}}+\frac{R_{1}}{R_{j+1}}\left(u_{j}\left(\frac{R_{j+1}}{R_{1}}\right)-\frac{1}{\epsilon }u_{j+1}\left(\frac{R_{j+1}}{R_{1}}\right)\right)\\ \; & \mathrm{for}j\mathrm{odd}, \end{array}$$
(12)$$\begin{array}{cc} {\frac{du_{j}\left(\rho \right)}{d\rho }|}_{\rho =\frac{R_{j+1}}{R_{1}}}= & {\epsilon \frac{du_{j+1}\left(\rho \right)}{d\rho }|}_{\rho =\frac{R_{j+1}}{R_{1}}}+\frac{R_{1}}{R_{j+1}}\left(u_{j}\left(\frac{R_{j+1}}{R_{1}}\right)-\epsilon u_{j+1}\left(\frac{R_{j+1}}{R_{1}}\right)\right)\\ \; & \mathrm{for}j\mathrm{even}, \end{array}$$
(13)$$u_{center}\left(0\right)=0$$

### 2.4. Properties of the Diffusion and Consumption Parameters

The dimensionless reaction-diffusion equations presented in Equations (6)–(9), introduce new dimensionless Michaelis-Menten parameters ν and κ, which were generated by dividing the old $V_{max}$ and $K_{M}$ parameters by the normalization constants $\left(D/R_{1}^{2}\right)*C_{0}$ and $C_{0}$, respectively. These new dimensionless Michaelis-Menten parameters combine both the consumption and diffusion parameters of oxygen.

In order to define the size of the nucleus we use values obtained from Huber (2007) [40], where the volume of the nucleus region $V_{nucleus}$ is 8% of the total cell volume $V_{1}$. In addition, the ER outer radius, $R_{3}$, is related to the nuclear radius, as observed in Lu (2009) [41], therefore we assume ER outer radius, $R_{3}$ to be approximately 1.9 times that of the nuclear radius, a value obtained from Lu (2009) [41]. The thickness of each membrane is chosen to be $\Delta r=0.5\%\cdot R_{1}$. The ER represented in Figure 2 is composed of multiple regions with membranes. The reaction-diffusion equations are solved assuming the regions cytosol, nucleus and a varying number of ER and separating membranes. The smallest number of region possible are 6 representing cytosol, nucleus, 1 ER region and 3 membranes. By increasing the number of ER regions to mimic the true scenario represented in Figure 1, we need to increase the number of ER. An increase in ER regions represents an increase in the membranes separating the various regions, such that for 2, 3 or 4 ER regions, we now have 8, 10 or 12 total regions, respectively.

### 2.5. Sample Preparation for Irradiation

The human urinary bladder cell carcinoma line T24 (ATCC, HTB-4) was purchased from ATCC and was cultured in McCoy’s 5A (Modified) medium (Cat.Nr.: 16600082, Thermo Fisher Scientific, Waltham, MA, USA) supplemented with 10% fetal bovine serum (FBS) (Cat.NR.:10082147, Thermo Fisher Scientific) and 1% HEPES buffer (1M) (Cat.Nr.: 15630056, Thermo Fisher Scientific) and 1% PenStrep (Cat.Nr.: 15140122, Thermo Fisher Scientific). Cells were cultured at both 1% and 21% atmosphere oxygen pressure in an InVivO2 400 hypoxic chamber (Baker Ruskinn, Sanford, MA, USA) and maintained at 37${}^{\circ }$ celsius, 5% CO${}_{2}$ and in a humidified atmosphere. Cell irradiation was carried out using a Multi Rad 225kV irradiator. Cells were initially synchronized (see next section for protocol) prior to irradiation with 6 Gy at room temperature. The cells that received 0 Gy (control) and 6 Gy were then plated into T25 flask to allow colony formation over 10 to 12 days. A total of 220 and 7500 cells were platted per flask, respectively for the 0 Gy and 6 Gy dose values. A total of 5 flasks were made per dose point. Cell survival was evaluated using a standard colony forming assay, where colonies were counted after 10 to 12 days.

### 2.6. Cell Cycle Synchronization

In order to synchronize cells in the late G1-phase, hydroxeaurea (HU) (Cat.Nr.: H8627, Sigma Aldrich, St. Louis, MO, USA) has been used as a synchronization agent, as shown by Ishida et al. [42]. A total of 2.5 × $10^{6}$ cells counted by Trypan blue exclusion method were seeded in a T175 flask (Cat.Nr.660175, Greiner Bio-one, Frickenhausen, Germany) in 2% FBS medium for 24 hours (h) . The day after, the cells were washed with Dulbecco’s Phosphate-buffered saline 1X (D-PBS) (Cat.Nr.:14190, Thermo fisher Scientific) and cultured in medium containing 0.5 mM hydroxyurea for a further 18 h . After synchronization, the cells were harvested at different time points (0, 2, 4, 6, 9, 12 h after releasing) to cover the cell cycle from the late G1-phase until the beginning of the G1-phase, where cell pellets were prepared for cell cycle and gene analysis. In addition, synchronized cells were irradiated using the Multi Rad 225 kV irradiator to 6 Gy.

### 2.7. Cell Cycle Analysis

Cell cycle analysis was performed by collecting $10^{6}$ cell from each time point and fixing them with 70% cold ethanol drop while gently vortexing. Samples were stored a +4 °C overnight and then processed for Propidium Iodide (PI) staining. For this purpose, cells were pelleted, washed with D-PBS, treated with 100 units/mL ribonuclease A for 30 min and, after washing, incubated for 30 min in a staining solution containing 20 μL/mL PI in PBS 1X. The resulting DNA content (i.e., the corresponding cell phase) was measured using FACSCanto II flow cytometer (Becton Dickson, Franklin Lakes, NJ, USA) and analyzed using FlowJo (10.5.0) software (TreeStar Inc., Ashland, OR, USA). A total of 70% of the cells in S-phase were considered to be synchronized.

### 2.8. RNA Extraction

For every time point, total RNA was extracted from $10^{6}$ cell with the High Pure RNA isolation kit (Cat.Nr.:11828665001, Roche Mannheim, Mannheim, Germany) according to the manufacturer’s instructions. DNAse-1 was used to purify the RNA from genomic DNA and the RNA amount and quality was then checked spectroscopically by means of a NanoDropND-100 (NanoDrop Technologies, Wilmington, DE, USA). Reverse transcription of 1 μg total RNA per sample was then performed using RT2 First Strand Kit (Cat.Nr.: 330404, Quiagen, Hilden, Germany ), according to the manufacturer’s instruction, in a StepOnePlus Real-time PCR system (Thermo Fisher Scientific). 20 ng of the synthesized cDNAs were amplified in 15 μL of a reaction mix consisting of nuclease free water, 20 pmol of each primer pair and 7.5 μL of Power SYBR green PCR Master Mix (Cat.Nr.:4357659, Thermo Fisher Scientific). The amplification conditions were as follows: 95 °C for 10 min (1 cycle), 95 °C for 15 s and 60 °C for 1 min (40 cycles).

### 2.9. Gene Expression with RT-qPCR Analysis

GAPDH was used as a housekeeping gene. GADPH is stable over the cell cycle [43], and can be used for any oxygen conditions and for T24 bladder cancer cells [44]. In the experimental data, the coefficient of variation (CV) was approximately 1.7 (average deviation of C${}_{T}$-values: 0.32) for the cell cycle. The averaging was performed using the CVs of 17 RT-qPCR plates (RT-qPCR, reverse transcription quantitative polymerase chain reaction). For comparing the stability of GADPH between synchronized and heterogeneous cell culture were checked. The CV was below 0.4% (average deviation 0.06, averaging the plate-CVs of 8 runs) and therefore GADPH can be considered as a suitable housekeeping gene). The RT-qPCR data were analyzed by using the comparative C${}_{T}$ method provided by Schmittgen et al. [45]. Accordingly, fold changes are defined by 2${}^{-\Delta \Delta C_{T}}$. Samples were run in duplicates and genes with a standard deviation below 0.05 in C${}_{T}$ values were considered to be significant. The C${}_{T}$ cutoff value was set at 36 cycles.

### 2.10. RT-qPCR Data Analysis of HIF and PHD Genes during Cell Cycle

The RT-qPCR studied involved studying the hypoxia inducible transcription factors (HIF) and prolyl hydroxylases (PHDs 1–3 in humans) genes during cell cycle and to correlate this variation with radiation response during the cell cycle, at both 1% and 21% oxygen pressure. The cellular radiation response is very dependent on the amount of oxygen present in the nucleus at the time of radiation, and therefore is an indirect assessment if the cell is hypoxic or not at the time of radiation. In addition, the hypoxia inducible transcription factors (HIF) are regulators of cellular responses to hypoxia in mammals [46]. In the present work, we evaluated the gene expression during cell cycle for the following genes: HIF-1α, HIF-3α, PHD1 and PHD2 at both 1% and 21% oxygen pressure. It is widely accepted that whereas HIF-1 and HIF-2 function as transcriptional activators, HIF-3 inhibits HIF-1/2α action. HIF-3α is degraded under normoxia. Under hypoxia or when overexpressed, HIF-3α binds to its target gene promoters and upregulates their expression. We then correlated gene expression with radiation response during cell cycle for 1% and 21% oxygen pressure, where the results presented within the manuscript are for 1% oxygen pressure because most in-vivo cancers have oxygen pressures between 0.2% and 2% oxygen pressure. By evaluating the HIF/PHD genes we are able to correlate cellular and nucleus oxygen concentration with radiation response of the cancer cells, which depends only on oxygen concentration of the nucleus.

## 3. Results

### 3.1. Characterizing Intracellular Oxygen as a Function of Michaelis-Menten Parameters, $V_{max}$ and $K_{M}$

We evaluated the intracellular oxygen concentration by solving the reaction-diffusion equations given in Equations (1)–(5) for ten (10) regions: Cytosol, ER, nucleus and 5 membranes. The results are presented in Figure 3 using dimensionless quantities (c.f. Section 2): (I) $O_{2}^{D}=\frac{O_{2}}{C_{0}}$, (II) $\rho =\frac{r}{R_{1}}$, (III) $\nu =\frac{V_{max}}{\left(D/R_{1}^{2}\right)*C_{0}}$ and (IV) $\kappa =\frac{K_{M}}{C_{0}}$, which represent the normalized intracellular oxygen, cellular radial distance and Michaelis-Menten parameters $V_{max}$ and $K_{M}$, respectively.

We assumed that the oxygen consumption rate is equal within the cytosol and ER ($V_{max,cyto}=V_{max,ER}$ and $K_{M,cyto}=K_{M,ER}$), with the following values for each $\nu_{cyto}=1,3,6$ and $\kappa_{cyto}=0.1,0.5,1.0$. The relative membrane permeability was set to ϵ=0.96 for the ER and ϵ=0.83 for the nuclear and plasma membranes. The lowest calculated value of $O_{2}^{D}$ within the nucleus was obtained for $\nu_{cyto}=6$ and $\kappa_{cyto}=0.1$, which yielded a hypoxic nucleus for external oxic conditions. We demonstrate that the highest consumption of oxygen within the cell was obtained with high values of $V_{max}$ and low values of $K_{M}$.

### 3.2. Characterization of Nuclear Oxygen as a Function of Extracellular Oxygen

We evaluated the nuclear oxygen concentration as a function of the extracellular matrix oxygen concentration $C_{0}$, for an increasing number of ER regions and membranes and corresponding to a total of 6, 8 and 10 regions. In addition, since the ER has a significantly higher level of ROS than the cytosol, we also evaluated how the nuclear oxygen concentration is affected by a higher removal of oxygen in the ER relative to the cytosol, where the values used were $\frac{\nu_{ER}}{\nu_{cyto}}=1.0,2.0,3.0,4.0,5.0$ and $K_{M,cyto}$ 3.1% atm for Human Neutrophils (HN) cells [27].

From Figure 4 some aspects of the oxygen level in the nucleus can be inferred. Firstly a general behaviour can be noticed: for small extracellular oxygen concentration (below 1% atm), there is a fast and linear decrease of nucleus oxygen concentration due to the oxygen consumption in the ER and cytosol. In addition, as we increase the ER oxygen consumption rate relative to cytosol, by making $\nu_{ER}>\nu_{cyto}$, we see a progressive decrease in the nuclear oxygen concentration. For $\nu_{ER}=2\cdot \nu_{cyto}$ the nuclear oxygen is approximately 11% atm (for 20% atm ECM), while for $\nu_{ER}=3\cdot \nu_{cyto}$ we now have 8% atm for nuclear oxygen. We also notice that there is a “saturation plateau” obtained when $\nu_{ER}>5\cdot \nu_{cyto}$, for high extracellular oxygen (above 4% atm ECM). This saturation behaviour has also been observed in Lai (1982) [47], which evaluated oxygen uptake by Chinese hamster ovary (CHO) cells using the electron spin resonance (ESR) closed-chamber method. In the Lai (1982) study, the rate of oxygen uptake was found to saturate for extracellular oxygen concentration above 5% atm, which is consistent with our work where saturation occurs above 4% atm for $\nu_{ER}>5\cdot \nu_{cyto}$. The “saturation plateau” is also observed experimentally in many radiation biology studies when studying oxygen effects during radiation known as oxygen enhancement ratio (OER) [48,49], which saturates above 4% atm.

### 3.3. Characterization of Nuclear Oxygen as a Function of Relative Membrane Permeability

This section focuses on studying the effect of the relative membrane permeability on the nuclear oxygen. In Figure 5 we present nucleus oxygen versus extracellular oxygen for five different combinations of ER and Nuclear relative membrane permeability values. In Figure 5 we calculated the curves for: $\nu_{cyto}=3$, $\frac{\nu_{ER}}{\nu_{cyto}}=5.5$, $K_{M,cyto}=0.55\%$ atm. We observe in Figure 5 that the “saturation plateau” observed previously in Figure 4, when $\nu_{ER}=5\cdot \nu_{cyto}$, has a small dependence on relative permeability parameter ϵ. However, the “saturation plateau” is more dependant on the condition $\nu_{ER}$ ≥ $\nu_{cyto}$, than on the ϵ value used.

### 3.4. Characterization of Nuclear Oxygen as a Function of Cell Cycle Phase

In this section the reaction-diffusion model is used to estimate nuclear oxygen concentration for different cell cycle phases. For cells to proliferate they must enter the cell cycle, a process known to be highly energy demanding and very tightly regulated. The cell cycle is one of the most important and energy consuming processes within the cell, which ultimately results in cell division and the inheritance of genetic information into the daughter cells. Oxygen is a fundamental ‘nutrient’ in energy production during cell cycle progression. Lai (1982) [47] evaluated oxygen uptake by Chinese hamster ovary (CHO) cells during the cell cycle using the electron spin resonance (ESR). They observed that the rate of oxygen uptake increased, by as much as 50%, during the cell cycle with minimum in the early G1-phase and maximum in the late S-phase, where values were, respectively, 3.0 × $10^{-7}$ and 4.4 × $10^{-7}$ per cell per second. In addition, Denekamp [51] reviewed 9 experimental studies that evaluated the radiosensitivity of different cells through the cell cycle. For the majority of the cell lines assessed the cells were shown to most resistant in late S, and most sensitive in mitosis (M) and G2 phases. The late S-phase was found to be approximately 2 to 4-times more resistant that the G1- and M-phases, where the difference in radiation response was mentioned to be potentially related to oxygen.

The results presented in Figure 6 represent the nucleus $O_{2}$ concentration for four cell types with published $K_{m}$ values: (i) Chinese hamster ovary, CHO ($K_{m}=0.55\%$ atm [52]), (ii) human embryonic kidney (HEK) ($K_{m}=16\%$ atm [27]), (iii) intact human neutrophils, HN ($K_{m}=3.1\%$ atm [27]) and (iv) cell free system from human neutrophils ($K_{m}=2.3\%$ atm [27]) and were calculated assuming an initial extracellular $O_{2}$ of 4% atm. The CHO cells are healthy cells that are commonly used in radiation biology studies due the well known radiation response and the low $K_{m}$ value of 0.5% atm.

The nuclear oxygen concentration was obtained using our reaction-diffusion equations with the following values for the parameters: $\nu_{cyto}=3$, ${\frac{\nu_{ER}}{\nu_{cyto}}|}_{G1-Phase}=4.0$, and ${\frac{\nu_{ER}}{\nu_{cyto}}|}_{S-Phase}=6.0$, which corresponds to 50% increase in oxygen consumption from G1- to S-phase, and with $K_{M,cyto}=16\%,3.1\%,2.3\%,0.55\%$ atm and assuming a total of 10 regions. The reaction-diffusion model predicts that the nuclear oxygen decreases with decreasing $K_{M,cyto}$ for the G1-phase, the nucleus oxygen goes from 3.1% atm to 1.0% atm respectively for $K_{M,cyto}$ changing from 16% to 0.55% atm. In addition, at very low values of $K_{M,cyto}=0.55\%$ atm it is possible to observe a clear difference between the nuclear oxygen concentration in G1-phase and in S-Phase where a saturation plateau appears. For $K_{M,cyto}$ above 1% atm the nuclear oxygen is approximately equal for both cell cycle phases with S-phase values always below G1-phase.

### 3.5. Cell Cycle Analysis of HIF and PHD Genes for Bladder Cancer Cells, T24

Radiation clonogenic assays during cell cycle have been done extensively as reviewed by Denekamp [51], where cell cycle irradiation was done at either 4 Gy or 6 Gy. In the present section, we repeat the cell cycle radiation clonogenic assay at both 1% and 21% oxygen atmosphere for bladder T24 cancer cells with single dose of 6 Gy, but in addition perform gene analysis during cell cycle, where the T24 cells were synchronized by an arrest-release technique. Using HU, cells were synchronized in the late G1 and released for both 1% and 21% oxygen pressure, which allowed the monitoring of cell cycle progression at different time points (0, 2, 4, 6, 9, 12 h after releasing). To confirm the synchronization, FACS analysis was performed on PI-stained samples for 21% oxygen pressure (see Figure 7, left).

Figure 7 shows the FACS analysis of the cell cycle synchronization (at 21% oxygen pressure) and the survival fraction of cells receiving 6 Gy at different cell cycle time points (at 1% oxygen pressure). The survival fraction during cell cycle was performed at both the 21% and 1% oxygen pressure, where results of both 21% and 1% were similar. Therefore we decided to only report in the manuscript the 1% oxygen pressure results, because it mimics closer the in-vivo oxygen pressure of most cancers. The PI-analysis demonstrated that the time points reflected enrichment for cells in S-phase ($t_{2}$–$t_{4}$/$t_{6}$), late S-phase ($t_{6}$), G2-M phase ($t_{6}$/$t_{9}$), and G0/G1 phase ($t_{1}2$). After 6 h from releasing, most of the population was found in late S; and after 9 h, cells were mostly in the G2/M with some re-entering G1-phase. Overall, cells were found to be synchronized for at least 12 h. The present cell cycle study was done at 1% oxygen atmosphere, however we show via the radiation clonogenic assay and the genes analysis, that the cells become hypoxic during the cell cycle (at 6 h after release), making the cells radiation resistant. In Figure 7 (right) it is possible to identify a peak in survival at 6 h, corresponding to the late S-phase of the cell cycle.

Cell cycle gene expression for HIF-1α/3α and PHD1/2 is represented in Figure 8 for 1% oxygen pressure, where the gene expression was similar for both the 1% and 21% oxygen pressure. Cell cycle dependency of HIF and PHD presented in Figure 8 mimic that of the radiation clonogenic (Figure 7, right), where maximum point is observed at the late S-phase/early G2-phase corresponding to 6 to 9 h after release from synchronization, at 1% oxygen pressure. In the case of HIF-1α and HIF-3α, the maximum observed gene expression was observed between 6 h to 9 h after release from synchronization. Under hypoxia or when overexpressed, HIF-3α binds to its target gene promoters and upregulates their expression. We observe that maximum value of HIF-3α is observed at 6 h after release, corresponding to the maximum resistance to radiation, at 1% oxygen pressure. At 6 h after release, the nuclear oxygen content is the lowest (and potentially hypoxic), making it significantly more resistant to radiation effects. While in the case of PHD1 and PHD2, the maximum gene expression was observed at 9 h after release from synchronization, at 1% oxygen pressure.

### 3.6. Nuclear Oxygen Enhancement Ratio, NucOER: Ratio of Nuclear Oxygen of G1-Phase Relative to S-Phase

The previous results show that the Michaelis-Menten parameter $K_{M}$ plays a very important role in the cellular consumption of oxygen and final oxygen concentration within the nucleus. We showed that the saturation of the nuclear oxygen is achievable by assuming that $\nu_{ER}>5\cdot \nu_{cyto}$, this corresponds to a much higher consumption rate of oxygen within the ER than the cytosol. In addition, the cell cycle dependency in oxygen was evaluated by changing the $\frac{\nu_{ER}}{\nu_{cyto}}$ value from 4 to 6, representing a 50% increase in oxygen consumption as the cell goes from early G1-phase to S-phase. In the present section we evaluated the nuclear oxygen enhancement ratio, NucOER, as function of the Michaelis-Menten parameter $K_{M,cyto}$, where we define the NucOER as the ratio of the nuclear oxygen concentration for G1-phase relative S-phase taken at 4% atm extracellular oxygen. The NucOER is an indicator of how much additional oxygen is present in the nucleus for the G1-phase relative to the S-phase. For example a NucOER = 2 indicates that the G1-phase has twice as much oxygen as S-phase in the nucleus. In the following Figure 9 we represent the NucOER as function of $K_{M,cyto}$, where ${\frac{\nu_{ER}}{\nu_{cyto}}|}_{G1-Phase}=4.0$ and ${\frac{\nu_{ER}}{\nu_{cyto}}|}_{S-Phase}=6.0$. In addition, we represent two additional curves the “NucOER − 10%” and “NucOER + 10%”, which correspond respectively to ${\frac{\nu_{ER}}{\nu_{cyto}}|}_{S-Phase}$ values of 5.4 and 6.6. These two curves represent cells that have either 10% higher or lower oxygen consumption at the S-phase.

From Figure 9 we observe that for $K_{M,cyto}$≥ 2% atm, the NucOER values are all less than 1.6, with the majority very close 1.0. This indicates that the nucleus oxygen concentration is not significantly affected by the cell cycle. However, as $K_{M,cyto}$ decreases below 2% atm, NucOER value starts to increase very rapidly with decreasing $K_{M,cyto}$. For example, at $K_{M,cyto}=0.5\%$ atm we observe that the NucOER is approximately 2.8 ±0.8, where the 0.8 value is calculated from the “NucOER ±10%” curves. A NucOER value of 2.8 indicates that the nuclear oxygen concentration can be anearly 3 times higher in the G1-phase relative to the S-phase. In Bird (1975) [53] the authors studied the radiation response of CHO cells during cell cycle. The radiation response of cells is very sensitive to the oxygen concentration present in the nucleus at the time of radiation. Bird (1975) found that the late S phase was 3 times more resistant to radiation. Similar results are presented in the cell cycle review paper by Denekamp (1986) [51]. We indicate that our NucOER results are consistent with those presented by both Bird (1975) [53] and Denekamp (1986) [51], since a lower nuclear oxygen concentration about 3 times of S-phase relative to G1-phase, would make the cell 3 times more resistant to radiation in the S-phase.

## 4. Discussion

Oxygen is one of the most abundant elements on earth and plays a vital role in the production of energy via the process of respiration in cells, which subsequently leads to the production of reactive oxygen species (ROS). Evidence suggests that importance of cholesterol, a key molecule for cellular membrane organization, as a strategy to restrain free oxygen diffusion [54]. Cholesterol concentration in specific cell systems suggests non-random distribution. The lens fiber of the eye and alveolar epithelia, both of which have direct contact with air, show high densities of cholesterol [55]. The membranes of lens fiber cells have been extensively studied to understand oxygen diffusion, because of the particularly high cholesterol content that ranges from 2 (lens cortex) to 4 (lens nucleus) cholesterol/phospholipid molar ratio [3]. In the case of human lung alveoli, the main structure of gas exchange, they present differential cholesterol function at each level of oxygen movement from the air into the body. Surfactant and respiratory epithelium are composed of a mixture of proteis (10%) and lipids (90%), which include phospholipids and cholesterol (5–10%). It constitutes the first membrane barrier between the air and respiratory system, preventing alveoli collapse during respiration by reducing surface tension. Different species show different concentrations in surfactant and cholesterol, with higher cholesterol:phospholipid proportion in reptiles and other species [56]. For the case of cancer, a potential mechanism that is common to many is the modulation of cholesterol and sphingolipid-rich membrane microdomains, commonly referred to as lipid rafts. Membrane rafts have been implicated in the development and progression of several cancers including prostate, breast, lung and colon cancer [57]. The cholesterol content of rafts and there localization is known to regulate molecular interactions, and hence the function, of signaling proteins involved in carcinogenesis, such as RAS, EGFR and HER2. In addition, oxygen also plays a critical role in both chemotherapy and radiation therapy of cancer, where both rely on augmenting ROS stress within the cell, achievable only if oxygen is present. The increased ROS produced during chemotherapy or radiation therapy, create several types of DNA damage that ultimately lead to cell death. In a recently published paper we show that inhibiting cholesterol can alter radiation response of breast cancer cells (MCF7), using a FDA approved compound that inhibits the gene diacylglycerol acyltransferase 2 (DGAT2) involved in the final step of the synthesis of triglycerides in which diacylglycerol is covalently bound to long chain fatty acyl-CoAs [58].

In the present work we demonstrated in Figure 3 that cells can modulate their intracellular oxygen concentration. The removal of intracellular oxygen can lead to low nuclear oxygen levels that are ideally suited for resisting both chemotherapy and radiation therapy. The highest consumption of oxygen within the cell was obtained with high values of $V_{max}$ and low values of $K_{M}$, which represents the Michaelis-Menten parameters. In addition, the cells ability to modulate its intracellular oxygen levels depended significantly on the amount cholesterol present in the cellular membranes. High levels of cholesterol within the membrane can slow the diffusion of oxygen across the membrane, reducing the amount of oxygen that reaches the nucleus per unit time.

We also investigated the nuclear oxygen concentration as a function of cell cycle, since oxygen is a fundamental ‘nutrient’ in energy production during cell cycle progression. Lai (1982) [47] observed experimentally that the rate of oxygen uptake increased, by as much as 50%, during the cell cycle with minimum in the early G1-phase and maximum in the late S-phase. We evaluated nuclear oxygen concentration assuming a 50% increase in oxygen consumption from G1-phase to S-phase, and found that the nuclear oxygen concentration is significantly lower for S-phase relative to G1-phase for low values of $K_{M,cyto}$. We demonstrated that Michaelis-Menten parameter $K_{M,cyto}$ plays a very important role in regulating the cellular oxygen consumption and its final oxygen concentration within the nucleus. For low values of $K_{M,cyto}$ (below 1% atm) the nuclear oxygen concentration of the G1-phase was approximately 3 times higher than the S-Phase, as indicated in Figure 9. In addition, we investigated in bladder T24 cancer cells the intra-cellular oxygen variation during cell cycle. We showed in Figure 7 and Figure 8 that during the cell cycle there is a time-point (at 6 h) where both radiation survival fraction and gene expression (HIF-1/-3α) reach there maximum value. At this time-point of 6 h, the nucleus of the cell becomes hypoxic, allowing it to become significantly radiation resistant and yielding an upregulation of HIF-1/-3α and PHD1/2, which play key roles in the cellular response to hypoxia. The HIF/PHD genes allowed us to correlate cellular and nucleus oxygen concentration with radiation response of the cancer cells. We demonstrated that maximum HIF-1α and HIF-3α gene expression occur at the points of highest radiation resistance during the cell cycle, therefore the cell is most radiation resistant when cell nucleus is hypoxic.

## Figures and Tables

**Figure 1 ijms-23-05077-f001:**
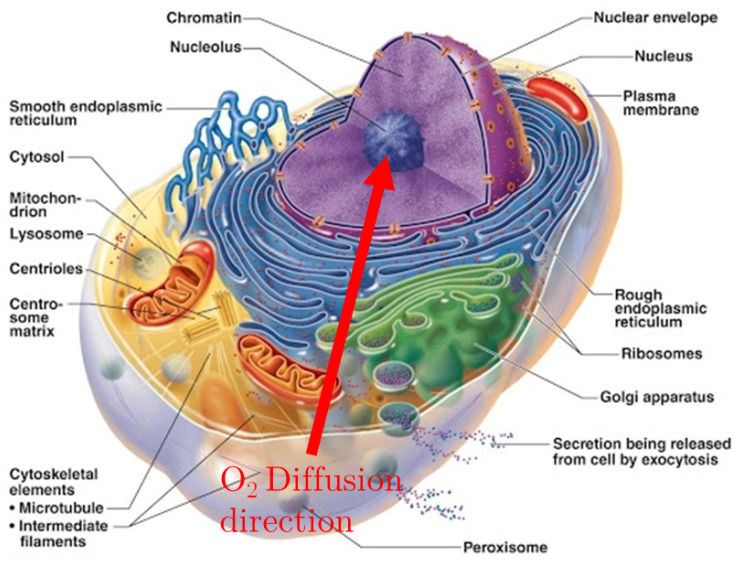
Section of the cell, showing the membrane’s development around the nucleus. The Red arrow shows the direction of Oxygen diffusion, from the extra-cellular matrix to the nucleus, crossing the ER membranes.

**Figure 2 ijms-23-05077-f002:**
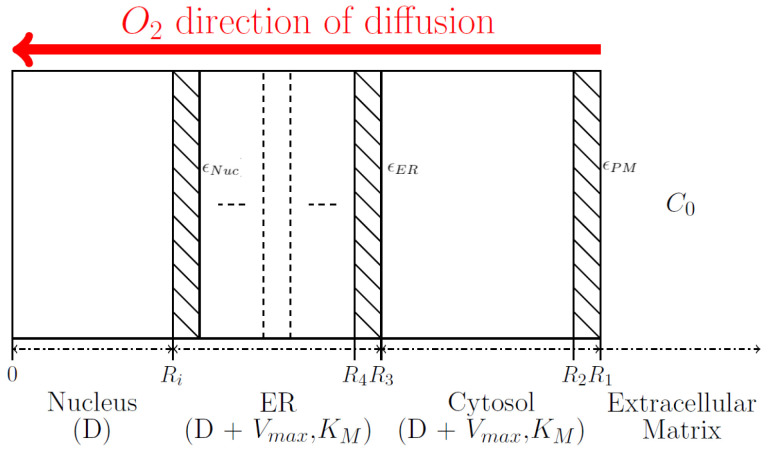
Representation of cellular regions. The cytosol region includes the mitochondria, while the ER includes also the Golgi apparatus. Regions with stripes represents the membrane interface. Note for ER, the number of membranes will be modified during the study.

**Figure 3 ijms-23-05077-f003:**
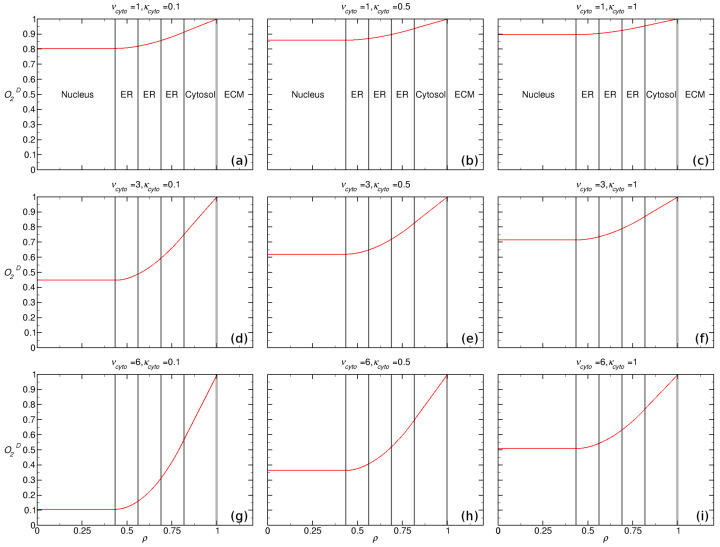
Intracellular oxygen concentration dependency on Michaelis-Menten parameters ($V_{max,cyto},K_{M,cyto}$) represented in normalized form $\rho =\frac{r}{R_{1}}$, $\nu =\frac{V_{max}}{\left(D/R_{1}^{2}\right)*C_{0}}$ and $\kappa =\frac{K_{M}}{C_{0}}$, considering 10 regions, $R_{1}$ the radial distance to outer cell membrane and $C_{0}$ is the constant ECM oxygen concentration surrounding the cell, where left-right $\kappa_{cyto}=0.1,0.5,1.0$ and top-down $\nu_{cyto}=1,3,6$.

**Figure 4 ijms-23-05077-f004:**
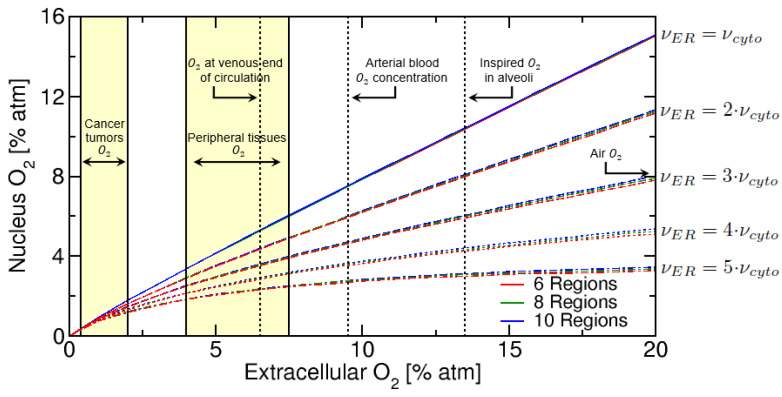
Nuclear oxygen versus extracellular oxygen, for 6, 8 and 10 total regions, representing an increasing number of ER regions and membranes. Oxygen vertical lines and yellow regions are taken from McKeown (2014) [50].

**Figure 5 ijms-23-05077-f005:**
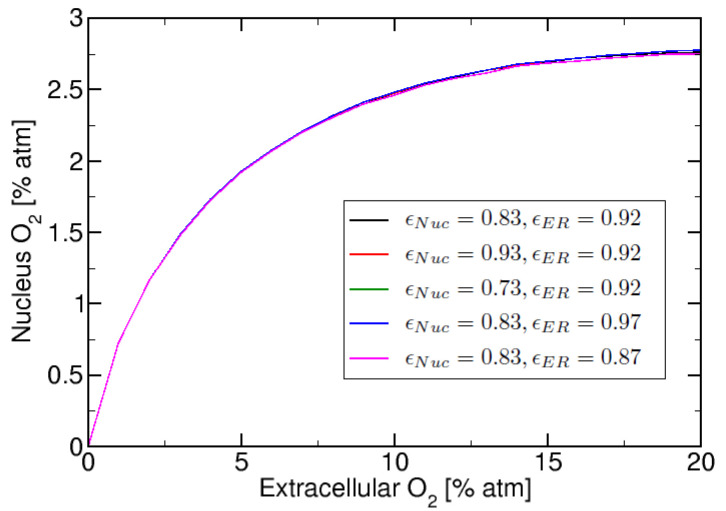
Nuclear oxygen vs. extracellular oxygen for 10 total regions (corresponding to cytosol, nucleus, 3 ER regions and 5 membranes) with varying combinations of ϵ values.

**Figure 6 ijms-23-05077-f006:**
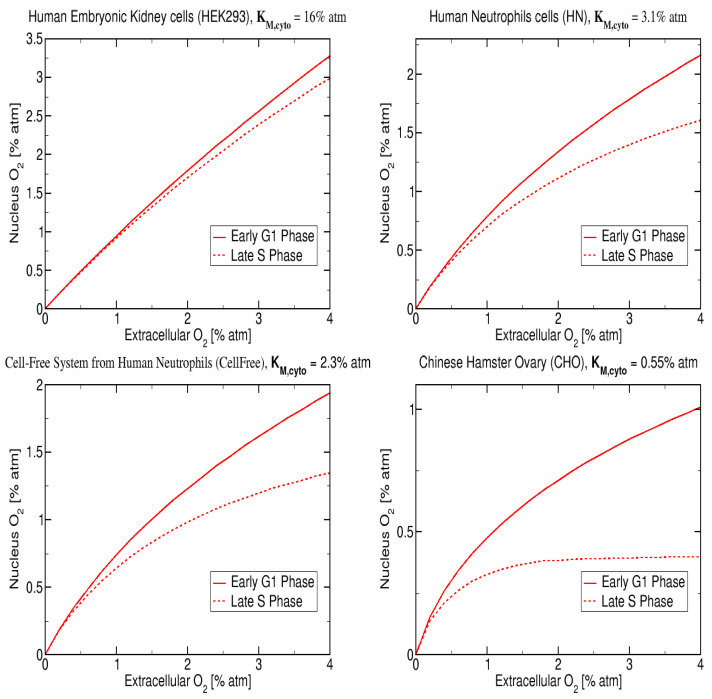
Model prediction for early G1 and late S phase and for four values of $K_{M,cyto}$: 16% atm, 3.1% atm, 2.3% atm, 0.55% atm and cell lines: HEK, intact human neutrophils (HN), cell free system from human neutrophils and CHO [27,52] and assuming initial extracellular $O_{2}$ of 4% atm.

**Figure 7 ijms-23-05077-f007:**
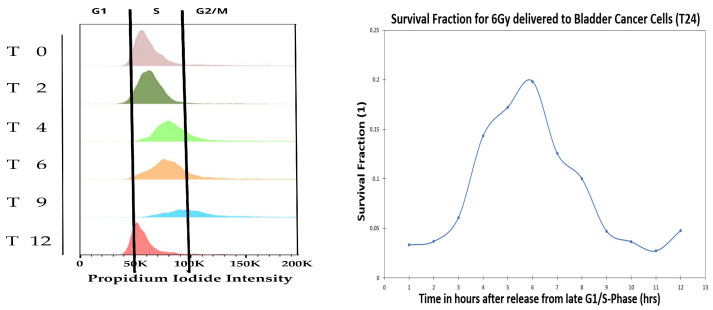
Fluorescence Activated Cell Sorting (FACS) is used to study DNA content during cell cycle using Propidium iodide (PI) at 0 Gy and 21% oxygen pressure. Survival fraction after 6 Gy delivery to the bladder cancer cells, T24, for 1% oxygen pressure.

**Figure 8 ijms-23-05077-f008:**
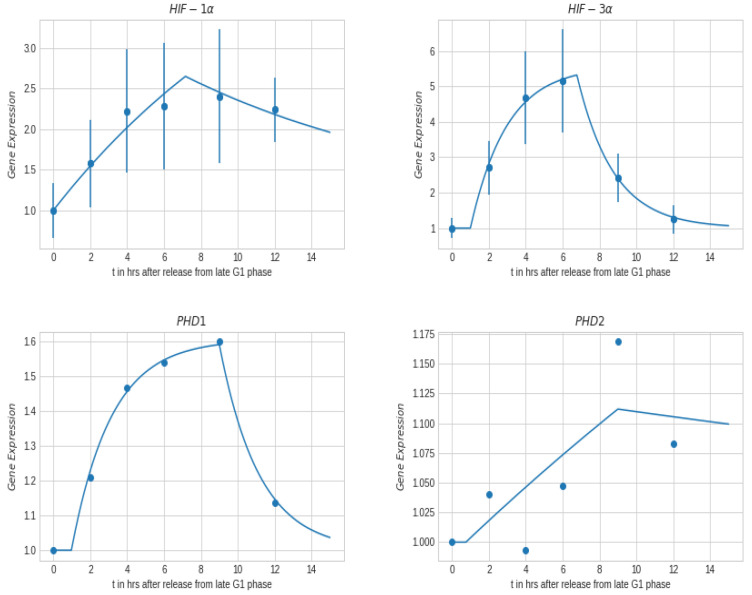
Gene expression obtained with RT-qPCR after release from cell synchronization at G1-/S-phase for bladder cancer cells, T24, for 1% oxygen pressure. (dots are the measured values, lines is best fit).

**Figure 9 ijms-23-05077-f009:**
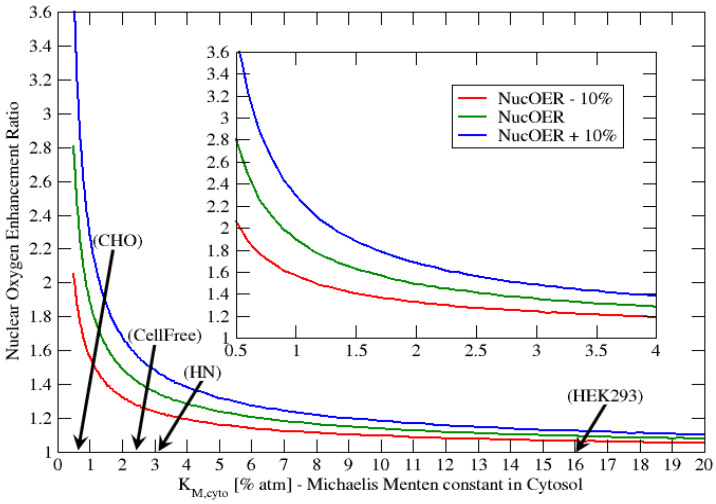
Concentration ratio vs. $K_{M,cyto}$ for fixed extracellular oxygen concentration $C_{0}=4\%$ atm and four different multiplying factors, taking as a reference $\nu_{ER}=1.5\cdot \nu_{cyto}$.

**Table 1 ijms-23-05077-t001:** Oxygen membrane permeability values at varying cholesterol levels from [5]. Relative permeability (ϵ) used for different organelles of the cell, which is defined as the ratio between the permeability values of a membrane containing cholesterol and one without cholesterol. * Interpolated value for ER membrane [24,25].

Cholesterol Content %	Permeability (cm/s)	Relative Permeability (ϵ)	ϵ of Organelle Membrane
0	52 ± 2	1	–
* 7.5	50	0.96	ER and Golgi App.
12.5	47 ± 2	0.9	–
25	48 ± 3	0.92	–
37.5	46 ± 2	0.88	–
50	43 ± 3	0.83	Nuclear and Plasma Membrane

## Data Availability

Not applicable.

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
