# Peer review of "Modulating Nucleus Oxygen Concentration by Altering Intramembrane Cholesterol Levels: Creating Hypoxic Nucleus in Oxic Conditions"

_ijms, 2022, doi:10.3390/ijms23095077_

Round 1

Reviewer 1 Report

The manuscript entitled “Modulating nucleus oxygen concentration by altering intramembrane cholesterol levels: creating hypoxic nucleus in oxic conditions” described studies examining a mechanism that cancer cells can modulate the oxygen concentration within the nucleus without the need of a hypoxic microenvironment. They found this ability to alter intra-cellular oxygen conditions depends on the amount of cholesterol present within the cellular membranes. This mechanism allows the cell to reduce the oxygen concentration of the nucleus, with implications to the Warburg Effect.

They also investigated the nuclear oxygen concentration as a function of cell cycle, and found that Michaelis-Menten parameter KM,cyto plays an important role in regulating the cellular oxygen consumption and its final oxygen concentration within the nucleus.

They demonstrate that during the S-phase cells can become hypoxic in the late S-phase/early G2-phase and therefore increase the radiation resistance.

The basic question is interesting and the data are consistent with most of the notion. And the paper is well written and discussed. I suggest this article can be accepted and published.

Author Response

I would like to thank the reviewer for his positive comments.

Reviewer 2 Report

The authors of this work evaluated the modulation of nuclear oxygen according to the cholesterol levels of cell membranes. The hypoxic condition has been extensively studied to understand the causes of radio- and chemoresistance, but this work would represent an important novelty and would be promising since the hypoxia has been often addressed by analyzing the tumor microenvironment and therefore considering aspects primarily related to growth, angiogenesis and to the irregular distribution of blood vessels rather than at intracellular level.

However, there are some inaccuracies that I suggest clarifying:

  1. I suggest reducing some of the concepts described in Materials and Methods that would be more appropriate for Discussion or Introduction. Likewise, some descriptions of the results would also be more appropriate to include in Discussion section. Overall, I would arrange the various sections avoiding repeating concepts that are redundant
  2. How many cells were seeded for the colony forming assay? How was the cell surviving fractions calculated? I suggest adding this information in Materials and Methods.
  3. No-irradiated controls (0 Gy) are missing in the survival curve (Figure 7) for each time points.
  4. I suggest clarifying the following points of Results:
  1. Why were the T24 cell cultured in 1% of oxygen pressure in hypoxic chamber and not in normoxia?
  2. So, why weren't the experiments with the T24 cell line conducted in parallel under normoxic conditions? This may by a critical point of the work since the results for clonogenic assay from cells cultured under normoxic conditions would be lacking; the same issue should be addressed for FACS analysis and for the evaluation of gene expression.
  3. After irradiation, were the cells replaced in hypoxic chamber or in normoxic condition for FACS, clonogenic and RT-qPCR analysis at the various time points?
  4. Why was a single dose of 6 Gy chosen rather than a dose response curve with more points?

5. I suggest clarifying the following points of Discussion:

  1. How is it possible to confirm that the acquisition of radioresistance during the cell cycle is due to the nuclear hypoxia condition related to the cholesterol changes and not to the hypoxic environment in which the cells were maintained for growth?
  2. Which experiment proves that intramembrane cholesterol levels reduce the amount of oxygen that reaches the nucleus per unit time?
  3. Do the authors propose future studies to evaluate and confirm changes in cholesterol?

Round 2

Reviewer 2 Report

I went carefully throughout the authors response to my concerns and unfortunately I found the manuscript little improved after revision. I strongly suggest a deep revision of the manuscript from a biological perspective. My main concerns are the following:

First and foremost, in the incipit of the abstract authors stated: “We propose a novel mechanism by which cancer cells can modulate the oxygen concentration within the nucleus, potentially creating low nuclear oxygen conditions without the need of an hypoxic microenvironment and suited for allowing cancer cells to resist chemo- and  radio-therapy

Authors stated that results of clonogenic assay, cell cycle and RT-qPCR are presented referring only to the 1% oxygen pressure, since these are “very similar” to those of the 21%. Such a statement is really peculiar. It is unlikely that hypoxic-related genes show superimposable profile either at 1% or 21% of O2 levels.

Second, if the authors want to demonstrate that the modulation of nuclear oxygen changes depend on the amount of cholesterol present within the cellular membranes rather than a hypoxic microenvironment, I still do not understand why the authors do not show the results obtained at 21% oxygen. These controls are of critical importance to confirm the experimental hypothesis of the study.

Third, authors stated that 1% O2 levels is representative of most of the cancer in in-vivo. Since this phenomenon can induce hypoxia within the nucleus, from this statement I do not understand how it was possible to demonstrate that the condition of nuclear hypoxia is not already induced by such low oxygen pressure regardless of cholesterol levels.

Forth, given the expertise on cholesterol lipid droplet biosynthesis impairment (doi: 10.3390/ijms221810102. PMID: 34576263; PMCID: PMC8466244) that the authors cited in the rebuttal and to fulfill the conceptual flaws of the manuscript, it is critical to perform a characterization of nuclear O2 in 21% O2 levels and impaired lipid droplet biosynthesis quantifying the levels of O2 within the cells, as main control experiment in the manuscript.

Finally, I would express at this stage, 2 additional major concern on experiments reported in figure 7 and 8.

Fig. 7: Graphs of PI intensity: it is not clear what gating strategy was used by the authors.

Fig. 7: Survival fraction: No standard deviation is reported. Neither statistical analysis was performed. As such, no conclusions can be based on the results herein presented.

Fig. 8: Here the authors show data from qRT-PCR that must be revised. i) No standard deviation is reported. Neither statistical analysis was performed. As such, no conclusions can be based on the results herein presented; ii) M&M: It is not clear what “Samples were run in duplicates and genes with a standard deviation below 0.05 in CT values were considered to be significant” means; note that values should be excluded performing specific tests; iii) It is not possible to perform statistical analysis on n=2 replicates per condition; iv) Axis should start from 0 and should have similar numbering interval (PHD2 graph gives a misleading information to the reader considering a min value of 1 and a max value of 1.17. v) Dots represent measured values and lines represent standard deviation. It is correct? If yes, why PHD1 and PHD2 were reported w/o?

At this point I do not feel confident to suggest this manuscript for publication.